

# Exploring the Greenland Ice Sheet's response to future atmospheric warming-threshold scenarios over 200 years

Alison Delhasse[1,*], Christoph Kittel[1,2,3,*], and Johanna Beckmann[4,5]

[1]Laboratoire de Climatologie et Topoclimatologie, Département de Géographie, UR SPHERES, ULiège, Liège, Belgium
[2]Physical geography research group, Department geography, Vrije Universiteit Brussel, Brussels, Belgium
[3]IGE, Univ. Grenoble Alpes, IRD/CNRS/INRAE/Grenoble INP, Grenoble, France
[4]Securing Antarctica's Environmental Future, Monash University, School of Earth, Atmosphere and Environment, Clayton, Australia
[5]Potsdam-Institute for Climate Impact Research (PIK), Member of the Leibniz Association, Potsdam, Germany
[*]These authors contributed equally to this work.

**Correspondence:** Alison Delhasse (alison.delhasse@uliege.be)

**Abstract.** The Greenland Ice Sheet (GrIS) plays a crucial role in sea level rise (SLR). We investigate its response to warming thresholds over two centuries using a coupled regional-atmospheric ice sheet model (MAR-PISM, respectively run at 25km and 4.5 km resolutions). We explore responses under global atmospheric warmings from +0.6°C to +5.8°C since pre-industrial temperatures and assess GrIS recovery if the climate reverts to present conditions, while prescribing unchanged ocean conditions. Our study then only evaluates the effect of atmospheric changes on the Greenland ice sheet. Moderate atmospheric warmings (+0.6°C to +1.4°C) yield steady and similar SLR contributions (from +8.35 to +9.55 cm in 2200), close to levels already committed under the present climate. Global temperature increases beyond +1.4°C mark a critical threshold, triggering non-linear mass loss due to feedback mechanisms like the melt-albedo effect and firn saturation. The SLR increase between the +1.4°C and +2.3°C experiments is larger (+7.56 cm), highlighting an accelerating mass loss. This trend is further reinforced by the even greater increase of 15.51 cm between +4.4°C and +5.2°C, underscoring the amplified impact of higher warming levels. Reversing the climate after surpassing +2.3°C demonstrates the potential for GrIS stabilization, though at a new reduced state of equilibrium (around 4% smaller). These findings underscore the impact of thresholds and time spent above them, highlighting the importance of limiting anthropogenic warming to maintain GrIS stability and mitigate long-term SLR.

## 1 Introduction

Melting of large land ice areas, such as the Greenland Ice Sheet (GrIS), will have a significant impact on sea levels. Recent higher temperatures, intensified by the Arctic amplification (Fettweis et al., 2017; Bevis et al., 2019), combined with lower cloud cover (Hofer et al., 2017), and blocking atmospheric events (Seo et al., 2015; Hanna et al., 2018; Tedesco and Fettweis, 2020; Hahn et al., 2020) result in a fourfold increase in the GrIS ice loss (Ryan et al., 2019; Noël et al., 2019) corresponding to a 13.6 mm sea level increase over 1990 to 2022 (Otosaka et al., 2023). Projections by the end of the 21st century suggest a strong dependence on future warming scenarios (Goelzer et al., 2020), while large uncertainties remain about persistent blocking atmospheric conditions (Delhasse et al., 2021) and their potential impact on increased mass loss (Beckmann and Winkelmann,



2023). In the longer term, and under scenarios of even higher temperatures, ice losses could become irreversible, ultimately causing the ice sheet to retreat (Aschwanden et al., 2019; Greve and Chambers, 2022).

The future of the GrIS is uncertain in the context of global warming. Various studies have attempted to determine the temperature threshold at which mass loss becomes irreversible, or whether it is possible to reach new stable states. On thousand-year timescales, a rise in global temperatures between +1.5°C and +2.3°C since the pre-industrial period could already lead to irreversible mass loss (Pattyn et al., 2018; Zeitz et al., 2022; Bochow et al., 2023). However, broader thresholds have been suggested, ranging from +1.5°C to +3.0°C (McKay et al., 2022), with higher thresholds such as +3.4°C identified in more a recent study (Petrini et al., 2025). Despite the existence of positive feedbacks such as the melt-elevation feedback, where melt thins the ice sheet, exposing it to higher air temperature, leading to enhanced melt, the complete disappearance of the ice sheet would require several (possibly tens of) millennia (Robinson et al., 2012). Furthermore, due to the inertia of the GrIS (Applegate et al., 2015; Gregory et al., 2020), it could be possible to exceed the maximum threshold temperature, provided that the climate cools quickly again (Ridley et al., 2010). This challenges the notion of a single threshold value, as the time scale of exposure to this threshold plays a crucial role in determining the ice sheet response.

All the previous studies overlook key processes affecting the Greenland Ice Sheet. While they rely on standalone ice sheet models that simulate ice sheet dynamics, they use low-resolution atmospheric forcings not representing interactions between changes in ice sheet geometry and the atmosphere, such as precipitation changes (Le clec'h et al., 2019). Additionally, their surface models, such as energy balance models (EBM) or positive degree-day (PDD) models, can introduce significant biases when estimating surface mass balance (Fettweis et al., 2020) and neglect the dynamical influence of a different geometry on winds and finally surface melt (Delhasse et al., 2024a). In contrast, polar-oriented regional climate models (RCMs) are run at higher resolutions than Earth System Models (ESMs), allowing for a more accurate representation of the ice sheet slope and its influence on precipitation or winds. They also incorporate advanced surface schemes that simulate snow and ice evolution more precisely, resulting in improved estimates of temperature and surface mass balance (Lenaerts et al., 2019; Fettweis et al., 2020). However, they lack the capability to simulate ice sheet dynamics, a limitation that can be addressed when coupled with an ice sheet model.

In this study, we therefore use a fully-coupled regional-atmospheric ice sheet model to evaluate the response of the GrIS to several atmospheric warming scenarios. We also question the state of the ice sheet on an idealized reversion to the current climate after exceeding a high level of atmospheric warming to assess the inertia capacity of the system. While using this fully-coupled model limits our experiments to a two-century time scale, it allows for insightful analysis over this shorter time-frame, yet policy-relevant and closer to climate change mitigation.

## 2   Methods

To evaluate the response of the GrIS to climate change and define its irreversible-melting temperature threshold, we perform several experiments using the polar-oriented regional atmospheric model Atmosphérique Régional Modèle (MAR v3.11.5, Fettweis et al., 2021; Delhasse et al., 2024a) coupled with the ice sheet model Parallel Ice sheet model (PISM, Khrulev et al.,



2023). Using MAR instead of a simpler energy balance or positive degree-day model requires more computational time but yields enhanced surface mass balance (SMB) estimations. Previous similar studies (e.g., Pattyn et al., 2018; Noël et al., 2021) neglect fundamental processes, such as the melt-albedo and melt-elevation feedbacks or the effects of changing slopes on precipitation. The coupling between MAR and PISM enables the representation of these processes, but is computationally more expensive. The coupling consists of updating the surface topography and ice mask calculated by PSIM in MAR and

updating the SMB and the surface temperature calculated by MAR in PISM each year while the atmospheric model has an internal time step of 180s and the ice sheet model has a monthly time step. We initiated the PISM experiments in 1961, applying 30 years of constant MAR forcing prior to the start of coupling in 1991. All coupling-specific details and the description of both models can be found in Delhasse et al. (2024a). The spatial resolutions of the models are the same (25 km for MAR and 4.5 km for PISM), as well as the model configurations described in this paper. We used a 4.5 km resolution, which reasonably captures

atmosphere-ice interactions while balancing computational efficiency, since higher resolutions primarily benefit ice–ocean interaction studies (Beckmann et al., 2019; Rückamp et al., 2020) not considered here (constant ocean conditions). Note that calving is represented using the von Mises calving law (Morlighem et al., 2016) enabling ice front to retreat if the stress is larger than the von Mises threshold of $1 \times 10^6$ Pa (Delhasse et al., 2024a). We do not account for Glacial Isostatic Adjustment, which is justified by the 200 year time-scale considered here (Zeitz et al., 2022).

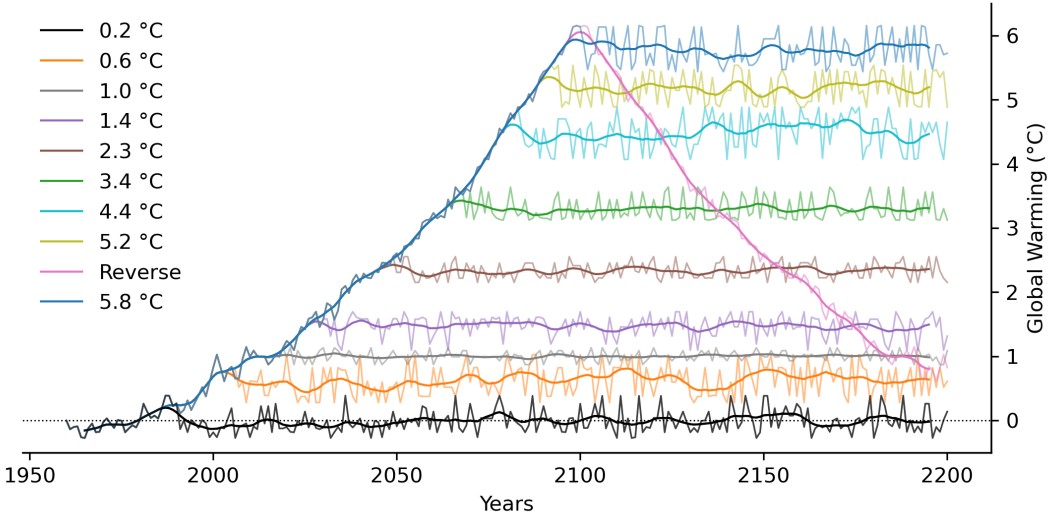

**Figure 1.** CESM2 global near-surface (2m temperature) warming (°C) compared to 1850-1950 (pre-industrial temperatures) for the 10 experiments of MAR-PISM coupling until 2200. Bold lines are 11-year running means.

The experiments rely on stabilized warming at nine atmospheric temperature thresholds from +0.2°C to +5.8°C compared to pre-industrial temperatures (1850-1950) (Fig. 1). The climate and different warming rates are derived from a Community Earth System Model version 2 (CESM2, Danabasoglu et al., 2020) simulation under the SSP5-8.5 scenario until 2100. We selected



CESM2 due to its high climate sensitivity, making it one of the most responsive models within the CMIP6 ensemble (the equilibrium climate sensitivity of CESM2 is +5.2°C compared to the CMIP6 ensemble mean of +3.2°C; Meehl et al. 2020). This choice enables to explore an extreme (very warm) future scenario for the Greenland Ice Sheet (GrIS) by 2100. To extend the projection, we randomly sampled the ten years until 2200 so that the warming is kept constant, but still with climate variability around this stable mean level of temperature increase. The ten-year periods, the exact resulting mean temperature, and the corresponding near-surface regional warming around Greenland are described in the Supplement (Table S1). The +5.8°C experiment corresponds to the 2-way coupled experiment analyzed in Delhasse et al. (2024a). We added a last experiment by reversing the large-scale climate forcing from 2100 to 2000 branching from the +5.8°C experiment in 2100 (hereafter the "reverse" experiment, in pink in Fig. 1). The year 2101 in our study aligns with the CESM2 forcing from 2099, while 2102 corresponds to 2098, and so forth. Consequently, the reverse year 2200 corresponds to climate conditions back in 2000. We considered the +0.2°C experiment as a control run as it corresponds to the reference climate (1961-1990) randomly sampled until 2200. We did not correct the contributions to sea level rise (SLR) by this control run, meaning that we kept the ice sheet model drift (+5.75cm SLR in 2200) highlighted by the +0.2°C experiment.

Finally, we integrated the surface mass balance components over the ice sheet surface area of each experiment (meaning that the mask varies) since this represents the exact area where the ice sheet can accumulate or lose mass. Using a common mask (the smallest extension of the ice sheet in all our experiments) would remove part of the ablation area artificially increasing the SMB for the low-warming simulations.

## 3   Results

Since the pre-industrial era, global mean air temperatures have already reached our threshold of 1.4°C (McCulloch et al., 2024) resulting in significant observed ice mass loss (Otosaka et al., 2023). Based on our results, global warming levels between +0.2°C and +1.4°C above pre-industrial temperatures correspond to a committed SLR of 8.35 to 9.55 cm in 2200. For comparison, extrapolating the observed mass loss rate of the Greenland Ice Sheet from 1980–2020 ($169 \pm 9$ Gt yr$^{-1}$,Otosaka et al., 2023) yields an equivalent sea level rise of $9.77 \pm 0.52$ cm by 2200, aligning with our results.

Ice mass loss increases with +2.3°C, and warmer experiments present a non-linear answer to stabilized level of temperature increase. A regional increase of 1°C in warming over Greenland, corresponding to global warming scenarios of +1.4°C to +2.3°C, leads to an additional 7.8 cm of sea level rise by 2200. In contrast, the same regional warming of 1°C, occurring between +4.4°C and +5.2°C global warming scenarios (representing less than 1°C of global warming difference), results in a more substantial sea level rise of 15.5 cm by 2200.

The reverse experiment also depicts a positive contribution in 2200 (Fig. 2A). However, unlike other warming thresholds, reversing the climate projected by CESM2 over 100 years stabilizes the GrIS mass loss at the end of our projection. After branching in 2100, mass balance is negative during 60 years (ie., positive sea level contribution, SLC) and positive afterward with a slightly negative SLC (Fig. S1). Although overshooting +2.3°C can lead to strong ice loss, our results suggest it is possible to stop the ice loss as long as this threshold has not been exceeded for too many decades of temperature increases





**Figure 2.** Projected A) contributions to sea level rise (cm), B) SMB (Gt/yr), C) runoff (Gt/yr), D) summer albedo (-), E) refreeze (Gt/yr, melt + rainfall - runoff), and F) mean elevation anomalies (m) compared to 1991, as simulated by MAR-PISM driven by different climate warmings of CESM2 by 2200. The thick lines represent the 21-year running mean (B-E).

beyond it. The GrIS could stabilize at a new equilibrium state with reduced ice thickness and surface area. However, this smaller configuration implies that ice mass loss—and the resulting sea level rise—is irreversible, at least within policy-relevant timescales.

As warming intensifies, the mass balance increasingly depends solely on the SMB, with progressively minimal contributions

from the dynamical processes (ice discharge). Over the present climate, the ice discharge is greater than the SMB in absolute terms leading to a negative mass balance (Fig. S1). In all experiments, the ice discharge decreases significantly, especially for experiments warmer than +4.4°C, where it becomes negligible compared to the surface mass balance-induced mass loss. The mass balance then roughly equals the SMB. Ice discharge is reduced due to the decrease in driving stress at the margins, which results from the stronger effect of margin thinning compared to the increase in slope; and the gradual inland retreat. Even in the





reverse experiment, where ice loss stabilizes, the SMB still dominates the mass balance, as the ice sheet does not re-expand. Additionally, we prescribe constant ocean forcings in all our experiments and do not simulate the effects of ocean warming on the ice dynamics, although the ice front can retreat following the von Mises law. Since this study focuses on the interaction between the atmosphere and the ice sheet, we will analyze SMB and atmosphere-ice sheet feedbacks in more detail to explain the Greenland Ice Sheet (GrIS) response, first for the threshold experiments and then for the reverse experiment.

## 3.1 Overshoot experiments

The SMB decreases non-linearly until 2100, with noticeable accelerations around 2025 and 2050 (Fig. 2A). After the climate stabilizes, the SMB continues to decline, but at a slower rate. Limiting the warming to 1.4°C strongly reduces the mass loss and keeps the GrIS in relative stability from now until 2200. The +0.6°C, +1°C, and +1.4°C experiments project a low decrease in SMB (from 120 Gt/yr to 210 Gt/yr compared to 1991) due to a limited increase in runoff, and then resulting in a relatively stable

(yet positive) SLC. Up to +1.4°C, the SMB, its components, as well as the ice sheet geometry (Fig. S2) are similar in 2200 to their branching state before 2050 (Fig. 2C and Fig. 3). A temperature increase larger than +2.3°C will cause a larger increase in runoff, reducing the SMB even further, resulting in ice geometries in 2200 much smaller than the +1.4°C ice geometries over the same horizon. At the climate stabilization around +2.3°C, the SMB is still positive (118 Gt/yr) but ends up becoming null before 2200. The transition between +1.4°C and +2.3°C is then an important threshold as at the ice sheet scale, losses

through runoff will exceed snowfall accumulation, leading to severe mass loss. Previous studies (Hofer et al., 2020; Noël et al., 2021) identified a threshold temperature at which the SMB becomes negative, marking the point where the ice sheet enters an inevitable decline. Beyond this threshold, mass loss becomes continuous, driven by surface ablation exceeding snowfall accumulation, further exacerbated by dynamic losses through ice discharge. However, these studies do not account for the long-term dynamics of the ice sheet nor represent the melt-elevation feedback. Our findings (also supported by Robinson et al.

2012; Petrini et al. 2025) indicate that, when considering both factors, this threshold is likely to occur at lower temperature increases than previously suggested.

A warmer stabilized climate does not result in a steady SMB for three main reasons: the melt-elevation feedback, the melt-albedo feedback, and the gradual saturation by liquid water of the firn. Ice melt thins the ice sheet, increases melt due to higher temperature associated with lower elevation, and strengthens again the ice sheet thinning. More melt also causes the

albedo to decrease from bright snow to darker bare ice albedo. This combines with the elevation feedback to strengthen melt. Finally, more melting saturates the firn, limiting the possibility of refreezing and leading to an increase in runoff, decreasing the SMB. Our projections highlight when these processes are triggered and suggest that they remain relatively weak until +1.4°C. Until this threshold, the decrease in SMB stays comparable to the committed change in SMB under the present climate. Once global temperature crosses the threshold, the non-linear response of the ice sheet to a regional warming of 1°C over Greenland

suggests strengthening of such feedbacks. These three potential feedback mechanisms and processes will be analyzed in the following paragraphs.

The mean albedo during summer exhibits an evolution similar to that of the SMB when stabilization of warming is applied: the albedo continues to decrease even under stable temperatures (Fig. 2D). As runoff continues to increase, more bare ice



areas with water at the surface are exposed, leading to a darkening of the global surface. For the experiments with +0.6 to
+2.3°C (i.e., 3°C of regional Greenland warming), the albedo decreases by only 0.01, whereas for the +3.4°C experiment (a
1°C increase in regional Greenland warming compared to the +2.3°C scenario), the albedo decreases by more than 0.02. A
sharp decline in albedo is projected around 2050-2075, marking the transition between the +3.4°C and +4.4°C experiments.
The +4.4°C experiment also exhibits a significant inter-annual variability, likely driven by large fluctuations in the melt areas
which are influenced by meltwater reaching flatter regions of the ice sheet. Since the minimum albedo for bare ice with liquid
water at the surface in MAR is fixed at 0.45, none of our experiments have reached the point at which this feedback would
cease during summer.

The response of refreezing differs across the various warming scenarios (Fig. 2E). For lower global warming levels (from
+0.6 to +3.4°C), when the regional temperature over Greenland rises by 1°C, refreezing increases consistently across the first
five experiments. However, for the +4.4°C and +5.2°C experiments, refreezing stops to increase when the warming is stabilized.
This suggests that the firn capacity is approaching saturation, as refreezing does not increase (or even begins to decrease) further
despite an ongoing rise in meltwater production (Fig. S3). For the +5.8°C experiment, the refreeze presents a peak well marked
around 2100, indicating that the firn has reached its capacity threshold. After this peak, refreezing declines rapidly before
stabilizing, as described in Noël et al. (2022). The refreezing capacity of the ice sheet firn mitigates the fast increase in runoff.
But once runoff exceeds a critical threshold and the firn maximum capacity is reached, this mitigation effect is lost. In our
experiment, the refreezing peak likely occurs between +4.4°C and +5.2°C, where the refreeze starts to decrease under a stable
climate while the meltwater production is still increasing. If the firn was not saturated, the refreeze should continue to increase
as projected for lower warming levels.

Finally, the melt-elevation feedback begins to play a more prominent role around 2100. The average thickness of the ice
sheet decreases across all our experiments, with an initial acceleration observed around 2050 (coinciding with the branching
of the +2.3°C experiment) and a much more pronounced acceleration occurring around 2090, between the +4.4°C and +5.2°C
experiments (Fig. 2F). However, the duration of our experiments (200 years) may not be sufficient to significantly melt the ice
sheet and lower its altitude enough to enhance this feedback. In our warmest experiment, this feedback contributes to 10% of
the total mass balance by 2200, as discussed in Delhasse et al. (2024a) (their Figure 4). This might suggest a stronger influence
of the melt albedo feedback and the progressive saturation of the firn (reduction in the mitigating effect on the increase in
runoff) to explain the persistent decrease in SMB and mass in warmer but stable climates by 2200.

## 3.2 Reverse experiment

As for the other experiments, the mass balance in the reverse experiments is directly linked to the SMB evolution. The surface
mass balance becomes positive again around 2160 and the mass balance twenty years later due to the ice discharge (Fig. S1).
This latest component remains stable from 2120 onward around -230 Gt/yr producing the time-delay before the transition to
the positive mass balance.

Reversing the climate results in a new equilibrium SMB for the GrIS, different from the one at the start of the 21st century.
As highlighted by Figure 2 and 3, the SMB becomes positive again around 2160, and remains lower at the end of the experiment




(see hereafter for the SMB hysteresis). Compared to 2000 when the climate forcing is the same, the runoff is higher in 2200 (403 Gt/yr over 2190-2200 vs 215 Gt/yr over 2000-2010). We found the reverse experiment runoff in 2200 to be higher than

the +0.6°C experiment runoff, whereas melt can be compared with the melt rates from +1°C and +1.4°C. The ice thickness decreases as long as the climate warms until 2100, but it is even lower with the reverse climate in 2200 than at the branching date in 2100 of this experiment as the mass balance is negative over most of the period (Fig. S2).

The thinning of the ice sheet and the time spent over the +2.3°C threshold are keys explaining the new equilibrium state in 2200. First, a thinner ice sheet leads to higher air temperatures in the reverse experiment, which explains the increased runoff

due to the higher surface melt as well as the greater amount of rainfall originating from snowfall that melts in the atmosphere due to warmer air temperatures. The 2200 reverse ice sheet is thinner than in the +3.4°C experiment but thicker than in the +4.4°C experiment. Compared to the +3.4°C experiment, the warmer climate during 90 years in the reverse experiment leads to more melt, potentially strengthening the melt-elevation feedback to also increase melt. Compared to 2000-2010, the melt is increased by 209 Gt/yr at the end of reverse experiment.

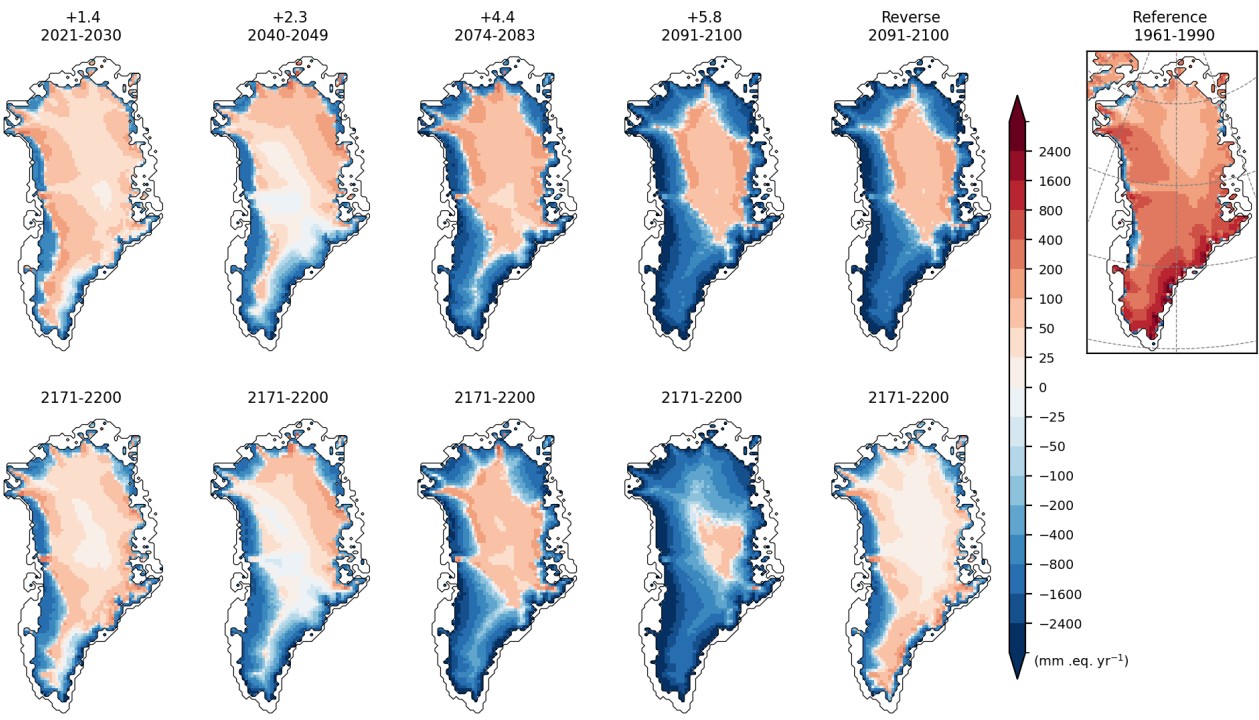

**Figure 3.** SMB anomalies compared to the reference period (1961-1990) on their respective evolutive ice masks. The first line exhibits anomalies of the 10 years selected to be repeated until 2200 with the climate of +1.4, +2.3, +4.4, and +5.8°C, as well as the last 10 years before reversing the climate for the reverse experiment. The second line exhibits anomalies of the last 10 years of the same experiments (2171-2200). The right panel is the SMB of the reference period (1961-1990).




However, our results underline the inertia of the system. The time spent above the +2.3°C threshold (around 110 years) coupled with the warmer atmosphere, and the fixed ocean conditions, is not long and warm enough to destabilize the ice sheet. Spending around 125 years at +4.4°C leads to larger ice mass loss than warming the climate until +5.8°C in 2100 before returning to the climate from 2000 (Fig. 2A). Even if the reverse experiment results in larger ice mass loss than the +2.3 and +3.4 experiments, its SMB in 2200 is higher suggesting a lower SLR contribution at a longer time-scale. Even with a thinner ice

sheet, the return to cooler conditions in the reverse experiment overrides the elevation feedback. The ice sheet has not thinned enough to continue to melt strongly under present climate conditions. In terms of irreversible mass loss over a short time range, the time spent over the temperature threshold seems then to be much more important than the amplitude of the imbalance. This again underlines the existence of stable overshoots. Although our simulations are relatively short (220 years) and the climate is still evolving, we believe the GrIS had sufficient time to approach a near-equilibrium state (at least for SMB), as evidenced

by the absence of a significant trend after 2150 ($p < 0.05$). This aligns with the findings of a previous study (Bochow et al., 2023), which demonstrated that the GrIS can achieve stability within 80 years. Therefore a warming of the atmosphere only up to +5.8°C in 100 years, followed by a return to the original atmospheric conditions in 100 years, can be seen as a stable overshoot.

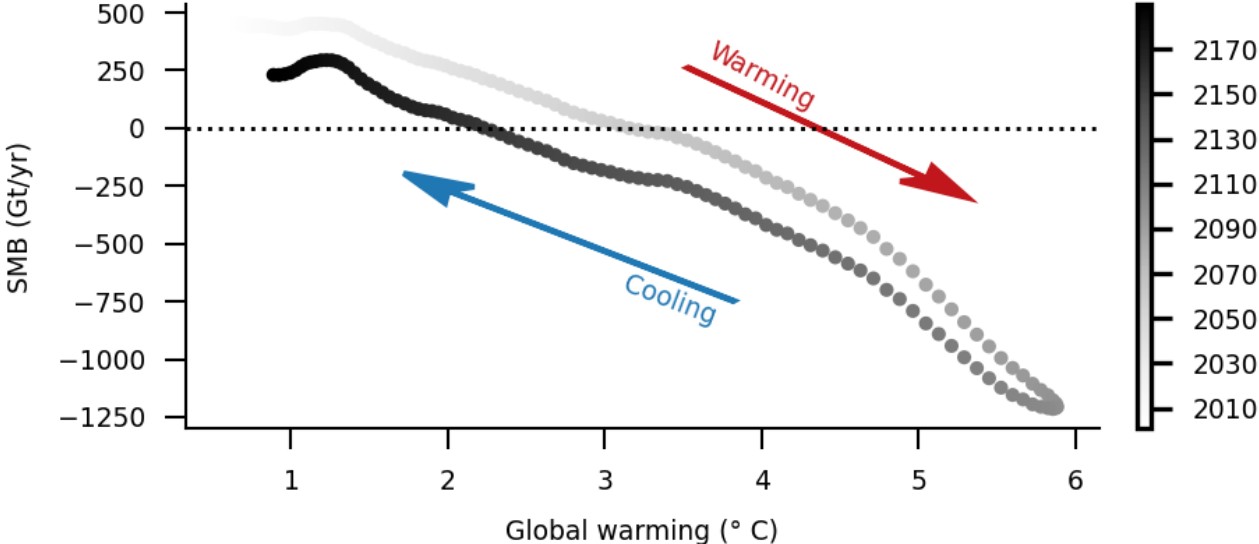

**Figure 4.** Projected SMB (Gt/yr) by MAR-PISM driven by CESM2 in the reverse experiment compared to the projected global warming of CESM2 colored by their years. A 21-year running mean is applied to the data. Red and blue arrows indicate respectively the direction of warming and cooling of the experiment.

Despite a lower SMB, the GrIS can reach an almost positive mass balance in the reverse experiment. Over the last 10 years of

simulations, the SMB is equal to 231 Gt/yr compared to 453 Gt/yr over 2000-2010. The runoff in 2200 highlights the saturation in liquid water of the firn. This limits the absorption of additional liquid water should the climate warm up again, leading to rapid ice losses as at the end of the experiment, when the SMB and mass balance rapidly decline again due to a few warmer



years. Ice discharge has decreased by 417 Gt/yr favored by the GrIS thinning, compensating the lower surface accumulation
(Fig. S1). This suggests that this new equilibrium state could be more stable in terms of mass balance (close to zero) than the
GrIS in its present state (negative).

The Greenland SMB has a hysteresis behavior linked to the ice sheet thickness and area. Figure 4 shows that the SMB does
not react the same way to warming and cooling in the reverse experiment. During the warming phase, the SMB decreases
when the warming is higher than +1.5C until the maximum atmospheric warming of +5.8°C to reach a value of -1250Gt/yr. In
the cooling phase, the SMB has a similar way to reincrease but with lower values (shifted by around -250 Gt/yr) and without
reaching back the values of the 1990s. The majority of new areas with negative SMB anomalies in 2100 show again positive
anomalies 100 years later in a colder climate. While the SMB becomes negative around +3.4°C in 2055, it only turns positive
again when the climate has only a +2.3°C warming. The lack of symmetry results from the lower albedo (Fig. 2D), higher
saturation of the firn (lower refreezing potential, Fig. 2E), the thinned ice sheet (Fig. 2F and Fig. S2) leading to higher melt
and runoff; as well as reduced ice sheet areas (Fig. S2).

## 225 4 Discussion

The experiments in this work highlight the impact of different rates of atmospheric warming as well as a gradual re-cooling
after a period of warming on the ice-atmosphere interface of the Greenland ice sheet. However, the seemingly equilibrium state
of the GrIS for the reverse experiment and in general all our results are likely conditioned by the absence of ocean warming
until 2100 and the coarse resolution of the ice sheet model.

The ocean response to the different warming rates applied here should increase calving that would amplify mass loss, but
the extent of this amplification remains uncertain (Fürst et al., 2015; Aschwanden et al., 2019; Choi et al., 2021). Such an
increase might be self-limiting due to the thinning and retreat of ice sheet margins, particularly since thinning enhances the
melt-elevation feedback and reinforces the predominance of SMB processes (Fürst et al., 2015; Aschwanden et al., 2019).
In the long term, retreat could enable the ocean to access deeper bedrock below sea level, potentially reactivating oceanic
feedback processes. However, this effect is counterbalanced by glacial isostatic adjustment, which is not represented in our
coupled model (Aschwanden et al., 2019). Several studies (Gillet-Chaulet et al., 2012; Goelzer et al., 2013; Fürst et al., 2015)
support the notion of a self-limiting increase in calving or highlight the greater contribution of surface processes (atmosphere
and SMB) compared to ocean-driven changes (Aschwanden et al., 2019). In contrast, Choi et al. (2021) suggests a more
significant role for calving in future ice loss. This divergence suggests that different outcomes are possible, with the potential
for destabilization as the ice sheet might lose more mass by 2100 before returning to present-day climate conditions in the
Reverse experiment.

Furthermore, the 4.5 km resolution of the ice sheet model in our experiments is likely too coarse to fully capture the dynamics
of outlet glaciers (Beckmann et al., 2019; Rückamp et al., 2020). However, even high resolutions (higher than 1 km), while
leading to greater projected mass losses, may not even be sufficient to represent the full complexity of ocean-ice interactions.
The absence of an imposed retreat mask associated with a coarser resolution in our study could compensate for the coarse



resolution (Rückamp et al., 2020). Finally, the coarse resolution of 4.5 km of the ice sheet model in this study reflects a balance between computational constraints and minimizes interpolation errors between the atmospheric and ice sheet model grids. The interpolation relies on gradients that are computed on the low-resolution atmospheric grid and applied on the higher-resolution grid of the ice sheet model. It assumes that these elevation gradients remain true even on the ice sheet model grid which might

not be true. Although higher resolutions for the ice sheet could improve the representation of certain ocean feedbacks, our setup reasonably captures the relevant dynamics to analyze large-scale interactions between the ice sheet and the atmosphere. Ice-atmosphere interactions, such as melt-elevation and melt-albedo feedbacks, might lead to larger mass losses than better capturing the ice-ocean interaction using a higher resolution: between +10-13% (this study and Zeitz et al. 2022), and between +12-47% for the melt-albedo feedback (Zeitz et al., 2022) compared to 5% for a higher resolution in the ice sheet model

(Rückamp et al., 2020). These comparisons underscore the importance of prioritizing the representation of ice–ice-atmosphere interactions in our study.

## 5 Conclusion

To determine the response of the GrIS to different warming thresholds, we performed coupled experiments with MAR and PISM. Compared with previous studies (Greve and Chambers, 2022; Zeitz et al., 2022; Bochow et al., 2023), we have analyzed

the effect of warming thresholds over periods of up to 200 years, closer to human time scales than the ice sheet time scale (thousands of years). Furthermore, the interactions between the atmosphere and the ice sheet dynamics are better taken into account through the representation of the firn evolution and positive feedbacks such as melt elevation and melt albedo.

Until 2200, not exceeding a global warming up to +1.4°C since the industrial revolution keeps the GrIS in relative stability while climates warmer than +2.3°C would lead to strong mass loss. This agrees with other studies finding stable GrIS up to

+1.5-+2°C (e.g., Bochow et al., 2023; Höning et al., 2023) and substantial loss for higher warming levels. While a cautious estimate of the warming threshold of sustained GrIS ice loss could be requested to determine the warming associated with a null SMB, we show that a positive SMB in +2.3°C can become negative leading to a sharp rise in sea level and the probable disappearance of the GrIS over longer time scales. The sustained GrIS ice loss is then likely to occur at a lower warming, particularly if we consider longer time scales (Robinson et al., 2012). This underlines the importance of the combined effect

of the length of time above the temperature threshold and the magnitude of the warming applied (Ritchie et al., 2021) the response of the GrIS being determined by both and not just one of the two elements. This aligns for instance with the findings of Gregory et al. (2020) or Bochow et al. (2023).

The exact determination of the warming threshold for sustained GrIS ice loss would require more short and long experiments with several warmer climates (including warmer ocean conditions), as well as several ice sheet and climate models to not

depend on the climate sensitivity of the model (as is the case in our study where we only used MAR and PISM coupled together). Furthermore, different climate forcing should also be taken into account, as some atmospheric circulation patterns favor greater increases in melting and could be only simulated by a few models (Delhasse et al., 2018, 2021). Further studies



should continue to analyze the importance of feedback mechanisms at various warming thresholds, as well as improve the representation of ice–ocean interactions, particularly by enhancing the resolution of the ice sheet model.

Finally, our study underlines the importance of remaining in climates colder than +2.3°C (in line with the objective of the Paris agreements). For instance, limiting the warming at +2.3°C without returning to a colder climate will not stabilize the GrIS and would even result in long-term mass loss. However, we demonstrate that it is possible to exceed this temperature threshold and, under certain conditions, to stabilize the GrIS in a new state of equilibrium.

*Code availability.* The data used for this study as well as both MAR and PISM models used to generate the data can be found in (Delhasse,
2023) and (Delhasse et al., 2024b).

*Author contributions.* AD and CK conceived the study. AD and JB performed the simulations. AD and CK wrote of the manuscript. All co-authors discussed the results, revised and contributed to the editing of the manuscript.

*Competing interests.* The authors have no competing interests.

*Acknowledgements.* Christoph Kittel is Postdoctoral Researcher of the Fonds de la Recherche Scientifique – FNRS. Computational resources
have been provided by the Consortium des Équipements de Calcul Intensif (CÉCI), funded by the Fonds de la Recherche Scientifique de Belgique (F.R.S.–FNRS) under grant no. 2.5020.11.



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
