# Peer review of "Exploring the Greenland Ice Sheet's response to future atmospheric warming-threshold scenarios over 200 years"

_EGUsphere, 2025_

## Community Comment (CC1)

This manuscript provides valuable insights into the response of the Greenland Ice Sheet (GrIS) to different warming thresholds using the coupled MAR-PISM model. The study effectively highlights the critical temperature thresholds (e.g., +1.4°C and +2.3°C) for GrIS stability and emphasizes the importance of both the magnitude and duration of warming. The results align with previous research and contribute important findings on GrIS behavior under future climate scenarios, with implications for sea level rise projections and policy decisions. The manuscript is well-structured and well-written. Pending minor revisions addressing the comments below, I support its publication.

**General comments**

While the study uses the coupled MAR-PISM framework to assess the GrIS response to warming thresholds, it lacks sufficient discussion on the added value of incorporating the MAR model. Although SMB processes are briefly addressed in Section 3.1, I would like to see deeper insights into how MAR contributes to projecting future GrIS SMB--especially given your finding that future SMB plays a more critical role than ice dynamics. Clarifying MAR's specific contribution would strengthen the manuscript and better support your conclusions.

In the Conclusion section, comparisons with other studies would be more appropriately placed in the Discussion. The Conclusion should focus more clearly on summarizing your key findings and highlighting the main insights regarding future GrIS responses to different warming scenarios.

**Specific comments**

L28-29: There seems to be a syntax issue with the phrase "in more a recent study." It should likely be "in a more recent study." Please revise for clarity.

L35: Replace "Greenland Ice Sheet (GrIS)" with simply "GrIS," as the full name has already been introduced earlier

L56: The term "enhanced SMB estimations" is ambiguous in this context. If you are referring to the MAR model's improved representation of SMB processes, please state this more precisely. As written, "enhanced" could be misinterpreted as implying increased SMB.

L68: Glacial Isostatic Adjustment ---> glacial isostatic adjustment

L75: Greenland Ice Sheet (GrIS) ---> GrIS

L76: The phrase "we randomly sampled the ten years until 2200" is unclear. Please specify the time period from which these ten years were randomly selected.

L84-85: The sentence "We did not correct the contributions to sea level rise (SLR) by this control run…" is somewhat confusing. As I understand it, you chose not to subtract the model drift (+5.75 cm by 2200 in the +0.2°C run) from the SLR estimates in other scenarios. If this interpretation is correct, I recommend rephrasing for clarity.

Page 5, Figure 2: I recommend repositioning the panel labels (A–F) from the y-axis labels to within each panel—preferably in a consistent location such as the upper right corner—for improved clarity.

L119: Greenland Ice Sheet (GrIS) ---> GrIS

L129: What do you mean null? zero or nan?

L131: Can you be more specific about this threshold temperature?

L140: change "darker bare ice albedo" to "darker bare ice"

L152: It would be helpful to quantify this change—how much is the albedo reduced, and over what area or time period

L154: Is the model assuming that if the ice sheet surface becomes flat, meltwater is stored locally without draining away? Is this behavior explicitly represented in MAR?

L177-178: surface mass balance ---> SMB

L178: "the mass balance twenty years later"? Do you mean that the mass balance becomes positive after 2180? Can you clarify this?

L182: Figure 2 and 3 ---> Figures 2 and 3

L211: "This limits the absorption of additional liquid water should the climate warm up again". Correct this sentence.

L241: Reverse ---> reverse

L275: forcing ---> forcings

L284: The phrase "under certain conditions" is too vague—could you specify what conditions are required for the GrIS to stabilize after exceeding the temperature threshold? Providing more detail would strengthen the conclusion.

---

## Author Comment (AC1)

Review of Delhasse, Kittel, and Beckmann: "Exploring the Greenland Ice Sheet's response to future warming-threshold scenarios over 200 years"

Dear Authors and dear Editor:

For transparency: I reviewed an earlier version of this manuscript submitted to a different journal. Since then, the manuscript has improved considerably and the current version addresses many of my previous comments.

Before I can recommend publication, I would like to ask the authors to address my comments below, most of which revolve around the writing quality and clarity.

Best,

Andy Aschwanden

University of Alaska Fairbanks

Dear Reviewer,

We would like to thank you for your comprehensive comments and suggestions, again on this version of the manuscript. We have incorporated your comments, which have significantly improved our paper.

Best,
Alison Delhasse, Christoph Kittel and Johanna Beckmann

(PS: you will find in red the suggested modification in our manuscript)

General Comments:

I recommend using acronyms sparingly as they make a manuscript less readable. In almost all cases, I find spelling it out improves the flow. Acronyms do have their place like for models like MAR or PISM or commonly used acronyms where the acronyms is better known that what it stands for (GPS, NASA, etc).

We retain the acronyms MAR, PISM, SLR, GrIS, SMB, and CESM2, while removing EBM PDD, ESM, CMIP6, and RCM as they are never used.

SMB and GrIS are commonly used in The Cryosphere. If the Reviewers and Editor think we should avoid using them, we will also remove it from our manuscript.

L 3: We explore responses to global atmospheric temperature increases from +0.6C to +5.8C since pre-industrial and…

Changed accordingly, thanks for the suggestion.

L 5: "Our study then only..." First, I'm not sure what the "only" refers to (what are you not evaluating?) and second, "then" is commonly used in constructs like "First, we did this, second, we did that, then we..." Please clarify.

We removed the "then" and rewrote the sentence.

Our study focuses exclusively on evaluating the effect of atmospheric changes on the Greenland Ice Sheet, without considering oceanic warming.

L 15: ...such as the Greenland Ice Sheet, are having…

Corrected.

L 18: "fourfold increase in ice loss" can you clarify whether or not this refers to rates of ice sheet mass change? Is it taken from table 2 in Otosaka 2023?

We weren't sure where this exact number came from, following Otosaka et al 2023 or Morlinghem et al., 2019, this should be more around sevenfold if we compare numbers from Table 2 in Otosaka 2023 (-35 Gt/yr in 1992-1996 vs -257 Gt/yr in 2017-2020) or -41 Gr/yr in 1990-2000 to -286 Gt/yr in 2010-2018 in Morlinghem et al., 2019. We

Melting of large land ice areas, such as the Greenland Ice Sheet (GrIS), is significantly impacting sea levels. Recent higher temperatures—amplified by Arctic amplification \citep{fettweis2017reconstructions,bevis2019accelerating}—along with reduced cloud cover \citep{hofer2017decreasing} and persistent atmospheric blocking events \citep{seo2015accelerated,hanna2018recent,tedesco2020unprecedented,hahn2020importance}, have led to an around sevenfold increase in GrIS mass loss when comparing rates from the 1980s to those observed between 2010 and 2020 \citep{mouginot2019forty,otosaka2023mass}.

L 19: "Mass loss projection for the end of the 21st century..."

Thanks for your suggestion. We'll make changes accordingly.

L 24: "causing the ice sheet to retreat". Please clarify. The ice sheet is already retreating, I assuming you mean that the Greenland Ice Sheet could disappear?

"In the longer term, and under scenarios of even higher temperatures, ice losses could become irreversible or cause the ice sheet's disappearance."

L 35-45 Paragraph requires some clarifications and disentanglement. Here is a suggestion based on what I think you are trying to say, modify and elaborate accordingly:

"All the previous studies overlook key processes..." I would change this to something like "Previous studies have a variety of shortcomings. Some studies employ standalone ice sheet models, ignoring, i.e., temperature and precipitation feedbacks between the ice sheet and the ice atmosphere (cite studies). Other studies use Earth System Models with interactive ice sheets (citations, e.g. Vizcaino 2015); while these models capture crucial feedbacks, their resolution of the atmosphere is often to coarse to resolve katabatic winds that impact surface melt, or their surface models can be biased because ...(elaborate). Polar-oriented regional climate models, by contrast, employ much higher resolutions than..., allowing for... However they lack the capability to simulate ice sheets, a limitation...

We tried to improve this paragraph with your suggestion keeping important informations about RCMs.

Previous studies have a variety of shortcomings. Some studies (Zeitz et al., 2022, Bochow et al., 2023) employ standalone ice sheet models, ignoring, i.e., temperature and precipitation feedbacks between the ice sheet and the ice atmosphere (Le et al., 2019). Other studies use Earth System Models with interactive ice sheets (e.g. Vizcaino et al., 2015, Muntjewerf et al., 2020); while these models capture crucial feedbacks, their resolution of the atmosphere is often to coarse to accurately resolve precipitation patterns associated with complex high-elevation topography (Fettweis et al., 2020). Polar-oriented regional climate models, by contrast, employ much higher resolutions than Earth System Models, allowing for a more accurate representation of the ice sheet slope and its influence on precipitation or winds influencing surface melt (Delhasse et al., 2024). They also incorporate advanced surface schemes that simulate snow and ice evolution more precisely, resulting in improved simulation of surface properties (Lenaerts et al., 2019, Fettweis et al., 2020). However, they lack the capability to simulate ice sheet dynamics, a limitation that can be addressed when coupled with an ice sheet model.

L 47: We also investigate the ice sheet's response to an idealized reversion to the current…

Corrected

L 54: ...coupled with the Parallel Ice Sheet Model (PISM, …)

Done.

L 55: "Using MAR instead of a simpler energy balance model..." Rephrase, maybe something                                                                                           like

"Using MAR to downscale coarse atmospheric forcing, combined with MAR's sophisticated surface energy balance model comes at a high(er?) computation cost, but results in a more realistic reproduction of surface mass balance (cite a study showing that).

Thanks, we will cite Fettweis et al., 2020 that compare different models to reproduce SMB over the Greenland Ice Sheet.

L 59: PSIM -> PISM

Thanks

L 61: "the ice sheet model has a monthly time step". PISM uses an adaptive time step by default. Did you enforce a 1 month time step?

Thank you for pointing this out. We indeed did not impose a fixed monthly time step in the simulations, and it appears that no specific limit for the maximum time step (`max_dt`) was set. We will revise the manuscript to clarify this.

"The ice sheet model used an adaptive time step to ensure numerical stability."

L 93-100. This is quite interesting and possibly worth a figure to visualize the non-linearity. How about plotting the sea level contribution (y axis) as a function of the per-degree increase in temperature (x axis)?

[Figure]

*Figure 1. Global atmospheric warming (°C) reached in each of the warming experiments in 2200 in function of the contribution to sea level rise (cm) of these experiments (black dots). In dotted red is the quadratic fit of this relation.*

The plot of this relationship illustrates pretty well our words, thanks for the comment. However, as we can already guess that's not a linear relationship in Figure 2A, we prefer to add this one in the Supplement for a seek of space and to not have multiple times the same information. However, if the editors or one of the reviewers think this figure should appear directly in the main manuscript, we remain open to include it.

L 109-10: "the mass balance increasingly depends on surface mass balance, with a progressively smaller contribution from ice discharge through the grounding line."

Changed accordingly.

L 128: clarify "over the same horizon"

We propose this new sentences.

A temperature increase above +2.3°C leads to a greater increase in runoff, further reducing the surface mass balance (SMB) and resulting in significantly smaller ice geometries by 2200 compared to those projected under a +1.4°C warming scenario.

L 132: "continuous" Do you mean "irreversible"

We're not sure irreversible is the right word (nor is continuous). We propose to change to "mass loss intensifies" as our reverse experiment shows that the melt can decrease in an idealized reversion to the current climate long after having exceeded the threshold.

L 134: What do you mean with "long term dynamics"?

We can remove *long-term* as these experiments do not account for ice dynamics at all. We also change the remaining part of the sentence:

However, these studies do not account for ice sheet dynamics or surface feedbacks such as the melt–elevation feedback.

L 142: "...remain relatively weak for global temperature increases below 1.4degC."

Thanks for the suggestion, we have changed accordingly.

L 165: "In our experiment" -> "In our experiments"

Changed accordingly.

L 179: "Ice discharge stabilizes around 2120 at about -230Gt/yr, causing the lag between SMB and mass balance turning positive.

Thanks for the suggestion, we have changed accordingly.

L 199: "Even if the reverse experiment has a larger mass loss in 2200 than the +2.3 and +3.4degC experiments, its SMB in 2200 is higher, suggesting a lower SLC on time scales longer than considered here."

Again thanks for this suggestion and you did before.

L 209: "The Greenland Ice Sheet's mass balance almost turns positive in 2200, despite the SMB being lower (231Gt/yr) compared to 2000--2010 (453Gt/yr)".

Changed.

L 210-15: Can you elaborate on this: "leading to rapid ice loss at the end of the experiment, when the SMB and mass balance rapidly decline again". I don't see this rapid decline in Figure 2 or S1. It appears to decline a bit, but to me this looks more like inter-annual variability. "Ice discharge has decreased by 417Gt/yr favored by the GrIS thinning, compensating the lower surface accumulation ". Do you mean "lower surface mass balance" instead of "surface accumulation"? Also "favored by ... thinning" sounds odd to me. Can you rephrase? "This suggests...could be more stable..." I'm afraid I'm not following here. The Greenland Ice Sheet is currently not in an equilibrium state. Could you clarify what you mean?

Thank you for the insightful comment. You're absolutely right—"rapid ice loss" as a sustained trend is misleading in this context. The drop in SMB and mass balance toward the end of the experiment primarily reflects interannual variability rather than a systematic decline. However, when comparing the beginning and end of the simulation, we observe increased sensitivity of the surface mass balance to warm years. This is due to long-term changes in surface snow and firn properties (e.g., reduced albedo, saturated firn limiting meltwater refreezing, and a lower ice surface elevation), which lead to enhanced runoff production even under similar atmospheric conditions.

Regarding the phrase "lower surface accumulation," you're correct—it should read "lower surface mass balance." As for "favored by the GrIS thinning," we agree the wording was unclear. We've revised it to better reflect the role of ice sheet thinning in reducing discharge.

Lastly, in saying the future GrIS "could be more stable," we meant that under the same atmospheric conditions, the Greenland Ice Sheet in the reverse experiment approaches a quasi-equilibrium state. This occurs because the reduced ice discharge resulting from long-term thinning compensates for the increased runoff, unlike the present-day GrIS, which is still adjusting and losing mass under current climate conditions.

Despite a lower SMB, the GrIS reaches an almost balanced state in the reverse experiment. Over the final decade of the simulation, the mean SMB is 231 Gt/yr, compared to 453 Gt/yr during 2000–2010. While the SMB appears variable near the end of the reverse experiment, this reflects interannual variability rather than a sustained decline. However, changes in firn and surface snow properties—such as reduced refreezing capacity due to firn saturation, lower albedo, and decreased surface elevation—lead to increased runoff potential during warm years, even under the present atmospheric conditions. Ice discharge has decreased by 417 Gt/yr compared to the present-day value, due to the long-term thinning of the ice sheet. This compensation between reduced ice discharge and enhanced runoff suggests that, under identical atmospheric conditions, the future GrIS in the reverse experiment may approach a quasi-equilibrium state, whereas the present-day GrIS is losing mass under the same forcing.

L 266: "With a cautious..." I don't understand this sentence. Could you please rephrase it?

We suggest to remove the first part of the sentence and rewrite as following

Determining a warming threshold for the disappearance of the GrIS based solely on the point at which SMB becomes zero is insufficient, as our results show that a positive SMB under +2.3 °C warming can later turn negative leading to a sharp rise in sea level and the potential long-term loss of the Greenland Ice Sheet.

L 285: fix citation types from \citep to \cite.

Done thanks.

---

## Author Comment (AC2)

This manuscript provides valuable insights into the response of the Greenland Ice Sheet (GrIS) to different warming thresholds using the coupled MAR-PISM model. The study effectively highlights the critical temperature thresholds (e.g., +1.4°C and +2.3°C) for GrIS stability and emphasizes the importance of both the magnitude and duration of warming. The results align with previous research and contribute important findings on GrIS behavior under future climate scenarios, with implications for sea level rise projections and policy decisions. The manuscript is well-structured and well-written. Pending minor revisions addressing the comments below, I support its publication.

Dear Reviewers,

We would like to thank you for your comments and suggestions. We have corrected all the mistakes and improved the conclusion section. See also our answer about how MAR contributes to projecting future GrIS SMB, especially compared to an Earth System Model. Best,
Alison Delhasse, Christoph Kittel and Johanna Beckmann

(PS: you will find in red the suggested modification in our manuscript)

**General comments**

While the study uses the coupled MAR-PISM framework to assess the GrIS response to warming thresholds, it lacks sufficient discussion on the added value of incorporating the MAR model. Although SMB processes are briefly addressed in Section 3.1, I would like to see deeper insights into how MAR contributes to projecting future GrIS SMB-- especially given your finding that future SMB plays a more critical role than ice dynamics. Clarifying MAR's specific contribution would strengthen the manuscript and better support your conclusions.

Numerous studies have investigated the future evolution of the surface mass balance (SMB) over Greenland using the MAR regional climate model (RCM), including Fettweis et al. (2013), Hofer et al. (2019), and more recently Glaude et al. (2021), as well as results from the ISMIP6 intercomparison project (Goelzer et al., 2020). Hofer et al. (2019) explored SMB evolution under various forcing scenarios. Delhasse et al. (2024) further demonstrated how coupling MAR with an ice sheet model affects future SMB evolution, in particular through negative feedbacks at the margins resulting from interactions between katabatic winds and changes in ice sheet geometry.

More relevant to your question, several key issues arise:

How does MAR simulate a different SMB compared to other models, including ESMs? To our knowledge, this comparison has not yet been systematically performed for Greenland under future scenarios. However, Fettweis et al. (2020) extensively discuss the advantages of using MAR or other RCMs over simpler models or Earth System Models (ESMs). Kittel et al. (2021) demonstrated the added value of MAR's higher spatial

resolution compared to CESM2, particularly for simulating precipitation in regions with complex topography—such as the Antarctic Peninsula or marginal valleys and ridges—highlighted in their Figure 11c–d. A similar improvement could be expected over Greenland's margins.

How does MAR compare to other RCMs in projecting future SMB? Glaude et al. (2021) addressed this by showing that MAR projects nearly twice the annual surface mass loss of RACMO by 2100 (−1735 Gt/yr vs. −964 Gt/yr), using the same CESM2 forcing as in our study. This discrepancy arises primarily from differences in runoff projections, which in MAR trigger a stronger melt–albedo feedback and a larger expansion of the modeled ablation zone.

What is the impact of this higher sensitivity on sea level projections? Goelzer et al. (2020) did not provide explicit RCM-related uncertainty estimates in their SMB-dominated projections of the GrIS, but noted that such uncertainties would likely propagate directly into the sea-level rise estimates. Following Glaude et al. (2025), MAR's greater sensitivity to warming tends to amplify SMB losses, potentially leading to higher projected melt contributions from the GrIS. It is therefore likely that our simulations represent a high estimate of SMB-driven contributions to ice loss, relative to other drivers such as ice dynamics and ocean interactions.

In the context of the H2020 PROTECT project, several ice sheet models have been forced with different RCMs under the same atmospheric boundary conditions. These coordinated efforts (Goelzer et al., pers. comm. 2025) will help quantify the model dependence and uncertainties associated with using MAR versus other RCMs. As this represents an extensive topic beyond the scope of our study, we propose to briefly address it in the discussion to contextualize and nuance our results better. This new paragraph also merges information that we moved from the Conclusion.

Although our results are likely influenced by using MAR and PISM, our results agree with other studies finding stable GrIS up to +1.5-+2°C (e.g. Bochow et al., 2023, Honing et al., 2023). The coupling between these two models enables a better representation of the interactions between the atmosphere and the ice sheet dynamics through the representation of the firn evolution and positive feedbacks such as melt-elevation and melt-albedo feedbacks. However, our simulations may be near the upper end of estimates of contributions for atmospheric–ice sheet feedbacks, relative to oceanic and dynamic ice losses. Glaude et al. (2024) showed that MAR projects nearly twice the annual surface mass loss of another RCM (RACMO) by 2100 (−1735 Gt/yr vs. −964 Gt/yr), using the same CESM2 atmospheric forcing. This difference stems from stronger melt–albedo feedbacks and a more pronounced expansion of the ablation zone in MAR. Since uncertainties in SMB forcing are expected to propagate almost directly into sea-level rise projections (Goelzer et al., 2020). While RCMs like MAR offer improved SMB representation over coarser models (Fettweis et al., 2020), future work should aim to quantify the influence of model choice on projections.

In the Conclusion section, comparisons with other studies would be more appropriately placed in the Discussion. The Conclusion should focus more clearly on summarizing your

key findings and highlighting the main insights regarding future GrIS responses to different warming scenarios.

Thanks for the suggestion. We remove the comparisons with other studies.

***Specific comments***

L28-29: There seems to be a syntax issue with the phrase "in more a recent study." It should likely be "in a more recent study." Please revise for clarity.
Indeed it should be "in a more recent study", corrected accordingly.

L35: Replace "Greenland Ice Sheet (GrIS)" with simply "GrIS," as the full name has already been introduced earlier
Corrected also for your comments below. Thanks for these corrections.

L56: The term "enhanced SMB estimations" is ambiguous in this context. If you are referring to the MAR model's improved representation of SMB processes, please state this more precisely. As written, "enhanced" could be misinterpreted as implying increased SMB.
You're right. Following the other reviewer's suggestion, we modified as:
but results in a more realistic reproduction of surface mass balance (SMB) estimations

L68: Glacial Isostatic Adjustment ---> glacial isostatic adjustment
Changed.

L75: Greenland Ice Sheet (GrIS) ---> GrIS
Done.

L76: The phrase "we randomly sampled the ten years until 2200" is unclear. Please specify the time period from which these ten years were randomly selected.
We completed the first sentence of the paragraph:
The experiments rely on stabilized warming at nine atmospheric temperature thresholds, ranging from +0.2 °C to +5.8 °C relative to pre-industrial levels (1850–1950), each defined based on a 10-year running mean of global temperature.
and now refer to the definition of the 10-year periods.
To extend the projections until 2200, we randomly sampled individual years from the 10-year period used to define each warming threshold, thereby maintaining a constant mean warming while preserving year-to-year climate variability.

L84-85: The sentence "We did not correct the contributions to sea level rise (SLR) by this control run…" is somewhat confusing. As I understand it, you chose not to subtract the model drift (+5.75 cm by 2200 in the +0.2°C run) from the SLR estimates in other scenarios. If this interpretation is correct, I recommend rephrasing for clarity.

Page 5, Figure 2: I recommend repositioning the panel labels (A–F) from the y-axis labels to within each panel—preferably in a consistent location such as the upper right corner—for improved clarity.
Done.

L119: Greenland Ice Sheet (GrIS) ---> GrIS
Done.

L129: What do you mean null? zero or nan?
Yes zero, we propose to improve the sentence:
[... ]but eventually declines to zero before 2200.

L131: Can you be more specific about this threshold temperature?
We propose to improve the paragraph:

Previous studies have identified either a timing—between 2046 and 2058 (Hofer et al., 2020, Noel et al., 2021)—or a threshold temperature (around +2.7 °C, Noel et al., 2021) at which the SMB becomes negative, marking the onset of irreversible ice sheet decline. Beyond these thresholds, mass loss intensifies, driven by surface ablation exceeding snowfall accumulation, further exacerbated by dynamic losses through ice discharge. However, these studies do not account for ice sheet dynamics or surface feedbacks such as the melt–elevation feedback. Our findings (also supported by Robinson et al., 2012; Petrini et al., 2025) indicate that, when considering both factors, the temperature threshold is likely to occur at lower levels of warming than previously suggested—implying that the onset of sustained mass loss could also occur earlier in time.

L140: change "darker bare ice albedo" to "darker bare ice"
Corrected as you suggested, thanks.

L152: It would be helpful to quantify this change—how much is the albedo reduced, and over what area or time period

We assume you're referring to: " A sharp decline in albedo is projected around 2050-2075, marking the transition between the +3.4°C and +4.4°C experiments." As numbers for the decline are already mentioned in the previous sentence (albedo step of 0.01 for experiments +0.6 to +2.3°C, and 0.02 for the next ones) to illustrate the changes in slope of this decline, we propose to improve the sentence as following:

As runoff continues to increase, more bare ice areas with water at the surface are exposed, leading to a darkening of the global surface. For the experiments with +0.6 to +2.3°C, the albedo decreases by only 0.01 per 1°C of regional warming, whereas for the +3.4°C experiment (a 1°C increase in regional Greenland warming compared to the +2.3°C scenario), the albedo decreases by more than 0.02. This larger decline in albedo is projected to occur between 2050 and 2075 if warming is not stopped before 2100

(experiment +5.8°C), amplifying the contrasts in albedo between the +3.4°C, +4.4°C, and +5.2°C experiments.

L154: Is the model assuming that if the ice sheet surface becomes flat, meltwater is stored locally without draining away? Is this behavior explicitly represented in MAR?

The horizontal flow of water is not explicitly represented in MAR yet. The sentence is confusing, we will modify it by:

The +4.4°C experiment also exhibits a significant inter-annual variability, likely driven by large fluctuations in the melt extent. As warming increases, the 0 °C isotherm rises to higher elevations where the ice sheet surface becomes flatter, resulting in larger inter-annual fluctuations in melt area.

L177-178: surface mass balance ---> SMB

Changed.

L178: "the mass balance twenty years later"? Do you mean that the mass balance becomes positive after 2180? Can you clarify this?

Yes, we clarified it by explicitly mentioning *2180* in place of *twenty years later*.

L182: Figure 2 and 3 ---> Figures 2 and 3

Thanks.

L211: "This limits the absorption of additional liquid water should the climate warm up again". Correct this sentence.

We changed the sentence while answering a comment to the other reviewer.

L241: Reverse ---> reverse

Done, thanks for this one and all the other similar mistakes.

L275: forcing ---> forcings

Thanks.

L284: The phrase "under certain conditions" is too vague—could you specify what conditions are required for the GrIS to stabilize after exceeding the temperature threshold? Providing more detail would strengthen the conclusion.

We suggest modifying the sentence to ensure consistency with our idealized reverse scenario, while maintaining a more generalized formulation appropriate for broader interpretation and applicability. Thanks for the suggestion.

However, we demonstrate that it is possible to exceed this temperature threshold and, by subsequently cooling the climate at a rate comparable to the preceding warming from 2100 onward, stabilize the GrIS in a new state of equilibrium.

---

## Author Response (AR2)

Dear Alison Delhasse and co-authors,

Thank you for thoroughly addressing the comments from the reviewers. I have only one further suggestion before recommending acceptance of your manuscript.

In the abstract, results, and conclusions, you describe the system as reaching a "new equilibrium." However, the evidence provided does not fully support this conclusion, as equilibrium implies not only stabilization of mass balance but also constancy in other dynamic variables such as ice velocity and related fields. To avoid overstatement, I recommend using a more cautious formulation, for example stating that "the mass loss has slowed down" or something similar, rather than implying a fully established new equilibrium. Please revise these sections accordingly and clarify your interpretation to ensure consistency with the presented results.

Best wishes,
Cheng Gong

Dear Editor,

Thanks for your remarks. We definitely consider it in our last version of our manuscript. We agree that our results and analyses do not allow us to state clearly that a new equilibrium has been reached. This is why we have replaced sentences containing this type of position with more nuanced statements, e.g., " close to equilibrium", "approaching a state of near-balance", or "slowdown in mass loss". These statements better reflect the potential towards a stabilization. Changes are mostly located in the abstract, results, and conclusion.

Best regards,
Alison Delhasse and co-authors